# Trade Sanctions and Agriculture Support in Milk and Dairy Industry: Case of Russia

**Mikhail Krivko [1],\*** and **Luboš Smutka [2]**

1   Department of Economics, Faculty of Economics and Management, Czech University of Life Sciences, 16500 Prague, Czech Republic

2   Department of Trade and Finance, Faculty of Economics and Management, Czech University of Life Sciences, 16500 Prague, Czech Republic; smutka@pef.czu.cz

\*   Correspondence: krivko@pef.czu.cz

**Abstract:** Economic sanctions between the European Union and Russia have significantly changed trade relations between them, while there are controversial assessments of sanctions' impact on both economies. Russian import ban has changed domestic producer prices in Russia, offering domestic producers a unique opportunity. There is an opinion that increasing self-sufficiency supports sustainable growth in agricultural production. At the same time, there is question of when and whether Russian import ban will be lifted? This paper offers an overview of changes in milk producer prices and support for milk producers in Russia in the period after the Russian import ban. We argue that currently the Russian Government has little incentive to lift import ban for milk and dairy products, as state support of agricultural producers has been decreased in significance for producers and was replaced by market prices support. Main findings suggest that all Russian federal regions experienced significant increases in transfers to producers from consumers; however, the pace of the increase appears to be different across regions. Paradoxically, the Western sanctions helped Russian milk and dairy industry to strengthen its position.

**Keywords:** consumer support estimate; import ban; milk and dairy industry; Russian Federation

## 1. Introduction

Milk and dairy industry occupy a very special position in the agriculture. According to Douphrate et al. [1], the major world's producers of milk are the European Union (about 25% share of the world dairy trade) and the United States (around 15%). International Dairy Federation [2] reports that the per capita milk consumption in the world increased from 102 to 110 kg from 2005 to 2014. The countries with the highest per capita consumption of liquid milk (above 90 kg) are Australia, Finland, and Sweden, while the United States, Canada, and Brazil yield between 60 and 90 kg per head, and the European Union and Russia, yielding from 30 to 60 kg in per capita consumption.

In 2014, following the events at the East of Ukraine, the United States and the EU imposed economic sanctions on the Russian Federation. Russia replied with counter sanctions imposing an import ban on agricultural goods from the EU, United States, Norway, Canada, and Australia, including meat, fish, cheese, milk, fresh fish, vegetables, and fruits. In this paper, we describe and assess the impact of sanctions on milk and dairy industry in Russian Federal regions. Overall, we argue that the sanctions do little harm to Russian agriculture but instead helped it to gain the new momentum and achieve positive changes, including the development of its dairy and milk sector and increasing the volume of inter-regional trade in milk and dairy products.

This paper is devoted to contributing to the discussion of when and whether the Russian import ban will be lifted. All in all, we argue that Russian Government has little incentive to lift the ban after five

years as state support of agricultural producers has been effectively replaced by market price support. While this support still represents taxpayers' money, prevalence of market price support over state support requires less administration costs and shows higher flexibility and resilience. Paradoxically, as Western sanctions were the main reason to introduce Russian import ban, sanctions helped Russian milk and dairy industry to catch the new momentum and to increase its efficiency.

Trends in milk and dairy industry of the Russian Federation significantly changed in the last decade. According to Reference [3], several important features of milk production and consumption emerged, especially the lag between production and demand, reduction in the number of livestock paired with increasing productivity, and increased competition with foreign manufacturers engaged in milk production in Russia. Guziy [4] also mentions the long investment cycles and low profitability level of dairy industry, as well as decreasing purchasing power of population. Growth of labor productivity has been also confirmed by Kharcheva et al. [5], who note that equilibrium level of salaries and productivity in Russian dairy industry has not been reached yet and salary does not fully stimulate increase in labor productivity. Competition with foreign producers which invest in production facilities in Russia has increased due to depreciation of ruble and therefore stronger position of foreign investors [6]. At the same time, Chinarov [7] points out that Russian domestic producers in 2016 had lower prices than farmers from Western Europe (39–43%) and Canada (53%). Nevertheless, there is an evidence of declining profitability of milk industry due to growth of costs outpacing growth of sale price of milk. He also revealed a shortage of domestic dairy products in the retail market to meet the paid demand and therefore suggests the introduction of state regulation in setting purchasing, wholesale, and retail prices for dairy products, insisting on stronger regulation and support for diary market. Interestingly, there is an evidence of relatively low price competitiveness of Russian agricultural producers due to real appreciation of the Russian ruble (a Russian currency denoted as RUB with a current exchange rate being slightly above 70 rubles for 1 euro (denoted as €)) as a result of difference in inflation rates in Russia and its main trade partners [8]. Looking on the problem from supply side, Petrick and Götz [9] demonstrated that current practices of subsidizing Russian agricultural enterprises have little influence to create herd growth. Authors also showed that controlling for herd size and other farm characteristics, individual farms received up to 18% higher subsidy amounts than enterprises, and subsidies seem to be more effective for individual farms in creating herd growth. However, absolute herd growth regressions suggest that the current subsidy policy does little to add cows.

Špička and Kontsevaya [10] identified several differences of milk processors in the Russian Federation and Central European countries in the period before Russian import ban. Russian companies show three times higher profitability (measured by return on assets, ROA, and return on capital employed, ROCE), two times lower stock turnover, and credit period shorter by 4 days. The authors suggested that the difference in profitability is primarily caused by higher inflation in Russia and capital structure of the enterprises.

Kharin [11] described an important feature of vertical price transmission on Russian diary market. On the example of Voronezh Oblast, the author has found that, during the period of 2002–2014, retail prices seemed to had influence on farm gate prices, while the opposite influence was rejected after testing. In the following paper on the matter of price transmission on the Russian dairy market, Kharin [12] investigated spatial (horizontal) price transmission in four Russian Federal Districts (Central Federal District, Volga Federal District, Northwestern Federal District and Southern Federal District) using Hansen-Seo technique to identify threshold effects in cointegration relationships of price pairs. It helped to reveal long-run relationship via threshold vector error correction model (TVECM) for all the price pairs in scope of the research. He also mentions the differences in results stemmed from linear and threshold VECM analysis, highlighting that linear VECM showed rather low degree of price pairs integration.

Developments in Russian milk industry are tightly connected with the topic of the import ban, that was introduced by Russian Government. In August 2014, the Russian Federation replied to

the European Union's sanctions with so-called Russian embargo which represents import ban on the several European agricultural products, including meat, fish, cheese, milk, fresh fish, vegetables, and fruits produced in the countries, which joined the EU restrictive measures to Russia. Import ban relates to the goods originated in the United States, European Union, Norway, Canada, and Australia.

Regarding domestic effects of the import ban, some authors point out that the results vary across sub-sectors, with increases in production of pork and decreases among dairy, beef, and fish producers [13].

Liefert et al. [14] attribute the changes in imports (average annual meat import was 40% lower in 2014–2016 compared to 2011–2013) to the import ban, while changes in exports were partially caused by production-enhancing favorable weather (grain exports were 50% higher in 2014–2016 then 2011–2013). As per Boulanger et al. [15], modeling the sanctions' situation using common general equilibrium (CGE) approach revealed that import ban should result in 5.7% decrease of milk price in Finland and 3.2% decrease of milk price in Lithuania, while price on the Russian domestic market should increase by 6.1%. Authors also anticipated increases of import of dairy products from Belarus and New Zealand by 202 million EUR for each country as a result of supplier substitution. This finding is also supported by Venkuviene and Masteikiene [16], who suggest that the dairy sector, followed by the meat sector, in Central and Eastern Europe (CEE) are the sectors most suffered from the Russian import ban. Czech Republic, Slovakia, Poland, and Hungary (Visegrad Group) had experienced milk price drop and decreasing competitiveness due to the Russian embargo, while elimination of milk quotas and cheap import products also played a significant role (see, e.g., Reference [17–19]). Fedoseeva and Herrmann [20] argue that, even though the Russian import ban has affected German exports, there are several other important events with relatively higher impact for Germany, such as the EU-wide pork ban imposed by Russia in the beginning of 2014.

Banse et al. [21] assessed several trade policy scenarios and has suggested that possible removal of the Russian import ban will affect Russian agricultural production only to a limited extent, while there will be no effect on the EU. At the same time, depending on how competitive Russian farmers become, creation of free trade area from Lisbon to Vladivostok would benefit farmers in the EU more than farmers in Russia. Authors used general equilibrium model MAGNET to estimate several trade policy scenarios for the period of 2017–2030, and the results suggest slight decrease of dairy production in Russia (−0.8% by year 2030) and increase of dairy imports by 23%. Removal of the import ban would lead to 3% decrease in dairy production, which authors attribute to increased competition with high quality imports of previously banned products.

Some of the researchers also point out to the decrease in revealed comparative advantage of EU exports to the Russian Federation, especially in terms of meat, milk, cheese, apples, and vegetables (see, e.g., Reference [22]). Tleubayev et al. [23] studied the changes on Russian dairy market on the example of cheese products and found evidence of increased integration of regional markets, as well as higher speed of price adjustment.

Authors that study effects of the Russian import ban also point out that growing self-sufficiency in food and seafood is the result of the ban [24,25]. At the same time, since the establishing of the Russian Food Security Policy in 2010, average per capita food consumption improved, although the poor consume much less [26].

Nevertheless, Loginov and Stepanyan [27] argue that current per capita consumption of milk in Russia is below the recommended nutrition levels. Therefore, there is a significant gap to fill in order to reach food security and self-sufficiency. Djuric et al. [28] used a regime-switching price transmission model in order to identify possible changes in the long-run equilibrium between the pig meat prices of Russia and its main non-EU trading partners. The results indicate the reduction of transaction costs in pig meat trade between Russia and its main non-EU trading partners, followed by the increase in transmission of price changes in the long run. At the same time, authors concluded that domestic consumers bear the biggest burden of the import ban impact.

Russian domestic policy, in terms of food security, is also linked to the national security of the state. In other words, country is vulnerable if it is not self-sufficient in food production. It is

important to study the Russian import ban in this context. From this perspective, introduction of the Russian import ban fits into the general picture. The import ban is one of the instruments (as well as higher tariffs or non-tariff regulations) to allow domestic producers to increase production capacities and be more prepared for international competition once the ban is lifted. In essence, the Russian import ban does not differ very much from the trade policies of many other countries in the world, which use similar measures to protect domestic food producers using tariffs, quality regulations, or other non-tariff measures.

All these facts bring one to the important question: has the import ban provided benefit or cost to the Russian state budget? There is an evidence of increased self-sufficiency of the Russian milk industry after import ban, but is it still beneficial for the Russian Government to continue the ban? Is the ban sustainable for agricultural market? Can the import ban be a reason to increase food security of the Russian agricultural sector? These questions have not been answered fully in the existing literature; therefore, the current research tried to add to the ongoing discussion by addressing the question of cost of self-sufficiency on the example of the Russian import ban and the willingness of Russian Government to lift the ban.

## 2. Materials and Methods

The cost of self-sufficiency, as well as import ban, impact can be assessed in different ways. For example, Gohin [29] uses equivalent variation of well-being to calculate the impact of the Russian import ban. Equivalent variation (EV) is calculated as follows:

$$EV = E\left(P^0, U^1\right) - E\left(P^0, U^0\right), \tag{1}$$

where $P^0$–vector of prices for the initial situation (before the ban); $E(P, U)$–household expenditure function; $U^0$, $U^1$–utility, before the ban and after the ban, respectively.

Despite being relatively straightforward in theory, in practice, this approach requires us to determine two functions (household expenditure and utility) which cannot be easily obtained from statistical sources. At the same time, this approach is subject to several underlying assumptions, such as small variations of endogenous variables $P$ and $U$, constant returns to scale, and Shepard's lemma, stating that, given the prices on the market, the consumer will buy a unique ideal amount of goods to obtain the maximum of utility with lowest cost.

Another approach to study the cost of the Russian import ban was used by Boulanger et al. [15]. Authors used computable general equilibrium (CGE) modeling employing GTAP model (described by Hertel) [30] to calculate the possible changes in trade and prices between countries of the European Union and the Russian Federation as a result of the import ban. Based on the estimated prices they also estimated the equivalent variation of well-being using equivalent variation, as in previously mentioned work.

Both given approaches are albeit theoretically elegant but deal with variables that are sometimes difficult to obtain from publicly available statistical sources. At the same time, these approaches rely on underlying assumptions of neoclassical economic theory. Another feature of both methodological solutions is that they give aggregate results for the country but do not provide an insight on what the actual changes in prices are, as well as the well-being of producers and consumers in individual regions of the country in question. While it might not be a significant issue for countries with rather small differences in regional economics, it might lead to imperfect conclusions for countries with significant heterogeneity of regional economics. The Russian Federation is an example of such a country.

Our research tried to estimate the cost of the Russian import ban for consumers and producers of milk in Russia by using prices and production volumes from publicly available databases, such as the Russian Statistical Office and Organization of Economic Cooperation and Development (OECD), employing the framework of the Producer Support Estimate/Consumer Support Estimate (PSE/CSE)

developed and supported by OECD. The PSE/CSE framework is further modified in order to recalculate country CSE estimate to regional CSE estimates.

Consumer Support Estimate (or CSE) is one of the indicators characterizing the amount of transfers to consumers of agricultural products as a result of the policies adopted in the country of interest. CSE was developed by OECD [31], together with other indicators, such as Producers Support Estimate (or PSE) and others, in order to evaluate the amount and direction of support to producers and consumers of agricultural commodities. CSE is calculated for a specific commodity on country level. This approach gives an overview of a transfers in a country on macro level, while it does not capture the differences between separate regions. It can be a significant restraint for the countries with heterogenous regional structure of economic development. For this paper, we attempted to estimate the differences in CSE between different regions (also called federal districts) of the Russian Federation by taking milk as an example of a product in order to capture the influence of the import ban introduced by the Russian Federation for specified commodities on transfers to/from consumers.

Heterogeneity of milk production volumes and producers' prices of milk between federal districts of the Russian Federation during the time period is tested by Student's two-sample *t*-test and two-way ANOVA, with years and federal districts assumed to be the sources of variation. Null hypothesis of equal mean between variable is assumed for both tests. Data used in the research originates from following sources: (i) OECD PSE Database: aggregate data for the Russian Federation on CSE estimate, producer prices, production volumes; and (ii) Federal State Statistic Service of Russian Federation (Rosstat), including the regional producer prices, production volumes, and regional population.

While the Russian Federation consists of more than 80 individual regions, each region is a part of a federal district. Current analysis focuses on federal districts of the Russian Federation instead of individual regions. Federal districts of the Russian Federation are mentioned in the text using abbreviations shown in Table 1.

**Table 1.** Federal districts of the Russian Federation.

| Federal District | Abbreviation |
|---|---|
| Central Federal District | CFD |
| North Western Federal District | NWFD |
| Southern Federal District | SFD |
| North Caucasian Federal District | NCFD |
| Volga Federal District | VFD |
| Ural Federal District | UFD |
| Siberian Federal District | SIBFD |
| Far Eastern Federal District | FEFD |

Our data consists of time series for 2010–2018, as data for 2018 is the most recent available data. Research focuses on milk, as it is one of the products that are in the import ban list and for which the Russian Federation is in the position of net importer. Other products that correspond to these two requirements are potatoes, beef and veal, pork, and poultry. Recalculation methodology, which is used in current paper, was applied for potatoes market in federal districts of the Russian Federation by Krivko et al. [32]. Descriptive statistics of the dataset are presented in the Table 2.

CSE for milk consists of two components: transfers to producers from consumers (TPC) and excess feed cost (EFC). As EFC is less than 5% of CSE for the time series in scope of current research, excess feed cost is omitted. As a result, CSE is equal to TPC. Therefore, Transfers to Producers from Consumers for milk is calculated as follows [31]:

$$TPC_c = QP_c \times RP_c - QP_c \times PP_c, \qquad (2)$$

where $QP_c$-production volume of commodity $c$; $PP_c$-producer price of commodity $c$; $RP_c$-reference price of commodity $c$.

**Table 2.** Descriptive statistics of the dataset.

| Federal District | Min | Max | Mean | Median | Std. Dev. |
|---|---|---|---|---|---|
| **Production Volumes** | | | | | |
| CFD | 5393.30 | 5784.20 | 5561.78 | 5507.00 | 161.50 |
| NWFD | 1684.70 | 1836.30 | 1762.28 | 1761.35 | 52.12 |
| SFD | 3263.60 | 3578.30 | 3367.85 | 3296.95 | 132.24 |
| NCFD | 2357.80 | 2795.50 | 2631.76 | 2657.80 | 145.12 |
| VFD | 9349.10 | 10408.50 | 9707.46 | 9486.90 | 389.29 |
| UFD | 1897.80 | 2096.20 | 2000.08 | 2011.00 | 84.61 |
| SIBFD | 4856.30 | 5725.80 | 5393.53 | 5387.80 | 270.61 |
| FEFD | 505.80 | 591.40 | 553.26 | 551.20 | 29.00 |
| Producers' Prices | | | | | |
| CFD | 14,949.60 | 25,766.49 | 19,841.31 | 20,047.27 | 4515.57 |
| NWFD | 12,563.71 | 24,464.37 | 19,228.16 | 18,999.67 | 4561.77 |
| SFD | 14,628.65 | 26,081.73 | 19,793.81 | 19,309.61 | 3959.49 |
| NCFD | 12,961.20 | 24,543.54 | 19,211.01 | 18,820.52 | 4624.00 |
| VFD | 12,344.97 | 27,238.95 | 18,867.38 | 18,810.85 | 5150.80 |
| UFD | 11,996.58 | 24,033.75 | 18,064.42 | 18,261.34 | 4511.55 |
| SIBFD | 13,413.66 | 23,913.67 | 18,588.59 | 18,102.11 | 3904.19 |
| FEFD | 12,697.39 | 29,341.01 | 18,471.43 | 17,978.85 | 5307.11 |

Reference price $RP_c$ represents possible border price of the milk. As milk is usually a non-tradable commodity, reference price is calculated based on the prices of dairy products as per PSE Manual (OECD [31]). In terms of the current study, milk reference price for Russia is taken from the OECD database [31].

We calculate regional TPC proportionally from country TPC, taking into consideration differences in production volumes and prices of products between regions (federal districts), as follows:

$$TPC_{cj} = \alpha_{1cj} \times QP_c \times \alpha_{3cj} \times RP_c - \alpha_{1cj} \times QP_c \times \alpha_{2cj} \times PP_c \tag{3}$$

where $\alpha_{1cj}$-regional coefficient for $QP_c$; $\alpha_{2cj}$-regional coefficient for $PP_c$; $\alpha_{3cj}$-regional coefficient for $RP_c$. $TPC_c$ in different regions can be calculated according to the differences in productions quantities, producer's prices, and reference prices (assuming both $PP_c$ and $RP_c$ are constants among regions):

$$\alpha_{1cj} = \frac{QP_{cj}}{QP_c}, \tag{4}$$

$$\alpha_{2cj} = \frac{PP_{cj}}{PP_c}, \tag{5}$$

$$\alpha_{3cj} = \frac{RP_{cj}}{RP_c}, \tag{6}$$

where $QP_{cj}$-production volume of commodity $c$ in federal district $j$; $PP_{cj}$-producer price of commodity $c$ in federal district $j$; $RP_{cj}$-reference price of commodity $c$ in federal district $j$.

In case there are no differences in quality between imported and domestically produced products, and no weight adjustment made, reference price is equal to border price:

$$RP_c = BP_c, \tag{7}$$

where $BP_c$–border price of commodity $c$.

As prices used in calculation are adjusted to the farm gate level, the costs of transportation of imported product to country's wholesale market increase reference price, while costs of transportation of domestically produced products to the wholesale market decrease reference price. Since reliable

data on transportation costs in both directions are difficult to obtain, these costs can be omitted in majority of cases as per OECD methodology. This assumption also brings to the conclusion that:

$$\alpha_{3cj} = 1; j \in (1; n), \tag{8}$$

where *n*–number of regions.

As $RP_{cj}$ is equal among regions, $TPC_{cj}$ is only dependent on production volume $QP_c$ adjusted by regional coefficient $\alpha_{1cj}$ and on producers' prices $PP_c$ adjusted by regional coefficient $\alpha_{2cj}$.

Another approach to recalculate TPC to regional level might be by using production volumes and producers' prices as variables, i.e., using $QP_{cj}$ instead of $QP_c$. However, we opted for using regional coefficients in order to receive values comparable to OECD country estimates.

## 3. Results

Overall, the Russian Federation can be considered a country with significant regional differences in terms of milk industry. There are several dimensions to this fact, including geographical and historical reasons. In terms of geographical dimension, the Russian Federation is the largest country in the world, covering the area of more than 17 million square kilometers, 16.3 million of which is rural land area, but only 13% of which is agricultural land. Thus, it is relatively difficult to consider the Russian Federation as a homogeneous economy because, unlike many other countries, differences in prices and production volumes between regions of Russia might differ more than between different countries of the world.

Market shares of federal districts have changed between 2013 and 2017 (Figure 1). Most remarkable changes are evident in two cases: market share of the Southern Federal District has risen from 10.82% in 2013 to 11.84% in 2017, while market share of the Central Federal District has increased from 18% in 2013 to 18.30% in 2017.

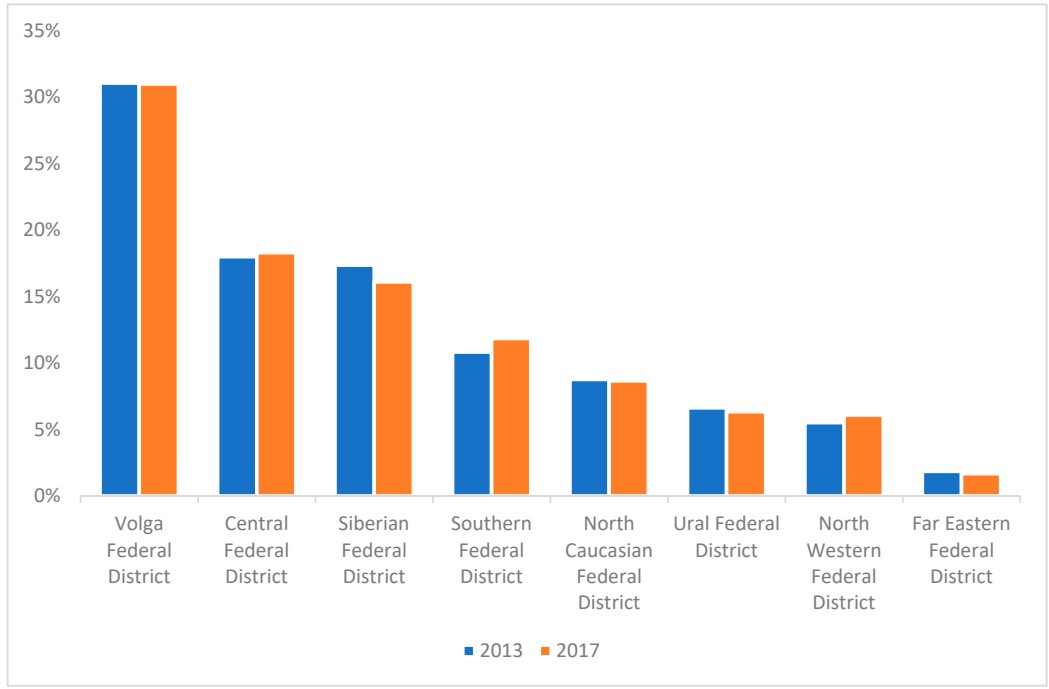

**Figure 1.** Changes in market shares of milk production of Russian federal districts in 2013 and 2017. Source: Organization of Economic Cooperation and Development (OECD) [31].

Market shares of federal districts presented in the Figure 1 show that federal districts are different in terms of production volumes, which is not unexpected; however, it is important to put into the context of changes induced by Russian import ban.

Heterogeneity of regional economies of the Russian Federation might be assessed by two means. Student's two-sample *t*-test can be used to test the null hypothesis of equal mean among production volumes of federal districts of Russia (see Table 3 below; abbreviations are the same as given in Table 1).

**Table 3.** *t*-statistics for milk production volumes in federal districts of the Russian Federation, 2000–2017. Source: own calculation based on Rosstat [33]. *** *p*-value < 0.01.

| | CFD | NWFD | SFD (+NCFD from 2010) | VFD | UFD | SIBFD | FEFD |
|---|---|---|---|---|---|---|---|
| CFD | - | 22.60 *** | - | 22.60 *** | - | 22.60 *** | - |
| NWFD | −22.60 *** | – | −20.41 *** | −69.61 *** | −3.26 *** | −50.00 *** | 29.92 *** |
| SFD (+NCFD from 2010) | −3.64 *** | 20.41 *** | - | −23.82 *** | 19.86 *** | −1.04 | 29.10 *** |
| VFD | 17.36 *** | 69.61 *** | 23.82 *** | - | 71.39 *** | 36.43 *** | 86.61 *** |
| UFD | −22.11 *** | 3.26 *** | −19.86 *** | −71.39 *** | - | −54.24 *** | 53.96 *** |
| SIBFD | −3.69 *** | 50.00 *** | 1.04 | −36.43 *** | 54.24 *** | - | 83.22 *** |
| FEFD | −30.14 *** | −29.92 *** | 29.10 *** | −86.61 *** | −53.96 *** | −83.22 *** | - |

Estimated *t*-statistics allow us to reject the null hypothesis of equal means in all the cases, except the pair of Siberian Federal District and Southern Federal District. The last pair shows similarity in mean values of production for the observable period. However, the *p*-value of the estimated statistics is higher than 0.1. Therefore, the null hypothesis of equal means cannot be rejected in this case. This allows us to conclude that Southern and Siberian FDs have common direction in production volumes. Nevertheless, all other regions do not show similarities in production volumes, suggesting high level of heterogeneity in terms of production among Russian regions.

Results of *t*-test for producers' prices of milk show that null hypothesis of equal means cannot be rejected in most of the cases, except pairs of Far Eastern FD and Volga, Ural, and Siberian FDs (Table 4). This fact shows that producer prices of milk in only one region (Far Eastern FD) did not have equal mean with other regions during the observable period. Comparison with production volumes suggests that regions of Russia are different in production volumes; however, producer prices follow a similar direction in most of the regions. Far Eastern FD takes specific place among all other regions in terms producer prices, but production volumes in this region follow a similar pattern.

**Table 4.** *t*-statistics for producers' prices of milk in federal districts of the Russian Federation, 2000–2017. Source: own calculation based on Rosstat [33]. *** *p*-value < 0.01.

| | CFD | NWFD | SFD (+NCFD from 2010) | VFD | UFD | SIBFD | FEFD |
|---|---|---|---|---|---|---|---|
| CFD | - | −0.04 | 0.16 | 0.64 | 0.27 | 0.48 | −2.14 |
| NWFD | 0.04 | – | 0.22 | 0.73 | 0.34 | 0.56 | −2.20 |
| SFD (+NCFD from 2010) | −0.16 | −0.22 | - | 0.50 | 0.11 | 0.32 | −2.37 |
| VFD | −0.64 | −0.73 | −0.50 | - | −0.41 | −0.21 | -2.87 *** |
| UFD | −0.27 | −0.34 | −0.11 | 0.41 | - | 0.22 | −2.56 *** |
| SIBFD | −0.48 | −0.56 | −0.32 | 0.21 | −0.22 | - | −2.81 *** |
| FEFD | 2.14 | 2.20 | 2.37 | 2.87 *** | 2.56 *** | 2.81 *** | - |

Another way to get a quick glance on the regional heterogeneity can be in the use of two-way ANOVA for milk prices and production data of Russian regions (Tables 5 and 6) with null hypothesis formulated as equality of means among regions and among years of observation. The dataset contains milk production volumes for years 2000–2017 and for 7 federal districts. Data for North Caucasian Federal District is included in the time series of Southern Federal District.

**Table 5.** Milk production volumes in federal districts of the Russian Federation, 2000–2017, ANOVA. Source: own calculation based on Rosstat [33].

| Source of Variation | SS | df | MS | F | *p*-Value | F Crit |
|---|---|---|---|---|---|---|
| Years | 2,466,179 | 17 | 145,069 | 0.674 | 0.82 | 1.724 |
| Federal Districts | 1,140,410,440 | 6 | 190,068,406 | 883.328 | 0.00 | 2.189 |

**Table 6.** Producers' prices of milk in federal districts of the Russian Federation, first differences, 2000–2017, ANOVA. Source: own calculation based on Rosstat [33].

| Source of Variation | SS | df | MS | F | *p*-Value | F Crit |
|---|---|---|---|---|---|---|
| Years | 59,037,585 | 13 | 4,541,352 | 1.390 | 0.18 | 1.848 |
| Federal Districts | 6,232,633 | 6 | 1,038,772 | 0.318 | 0.92 | 2.217 |

The null hypothesis for the ANOVA comprises that variances are equal between datasets. When federal district is the source of variation, the F-statistic is much higher than critical F-value; therefore, null hypothesis should be rejected. F-statistic has a high *p*-value when years are set as source of variation; therefore, F-statistic is not significant in this case and cannot be considered.

Analysis of variances for milk prices dataset shows that means are equal in terms of years and federal districts. Null hypothesis cannot be rejected, which leads to a conclusion of low heterogeneity of Russian regions in terms of milk prices. Based on this, it is difficult to assess transfers from consumer to producers as a result of the import ban on the country level. Deeper separation of the country into regions might show a more realistic picture of transfers and reveal the position of producers and consumers of milk after the introduction of the Russian import ban.

There is an evidence of significant differences between Russian federal districts in terms of both direction and value of transfers from consumers to producers of milk. At the same time, direction of transfers varies between federal and regional levels. Table 7 shows the TPCs for different commodities provided by OECD. In comparison to other commodities in the scope of the Russian import ban, milk has the highest value of transfers to producers from consumers.

**Table 7.** TPCs of selected commodities for the Russian Federation, 2010–2018, million RUB. Source: OECD [31].

| Year | Milk | Potatoes | Beef and Veal | Pig | Poultry |
|---|---|---|---|---|---|
| 2010 | 63,517.14 | 0.00 | 45,824.32 | 118,209.24 | 41,751.06 |
| 2011 | 82,757.29 | 0.00 | 21,711.82 | 88,386.28 | 34,095.87 |
| 2012 | 62,821.85 | 0.00 | 42,841.15 | 83,853.05 | 16,887.30 |
| 2013 | 1856.53 | 0.00 | 77,720.99 | 69,182.44 | 45,895.96 |
| 2014 | 56,648.11 | 26,961.84 | 39,001.33 | 133,733.21 | 2574.61 |
| 2015 | 121,054.12 | 0.00 | 69,209.71 | 78,672.39 | 27,442.29 |
| 2016 | 223,313.25 | 0.00 | 45,272.71 | 60,264.43 | 17,144.58 |
| 2017 | 143,685.30 | 0.00 | 42,419.05 | 59,222.22 | 54,052.25 |
| 2018 | 224,804.51 | 0.00 | 56,959.16 | 39,337.24 | 33,473.07 |

Data on TPC provided by OECD suggests that transfers to producers from consumers have significantly risen after 2014 in the case of milk and poultry; at the same time, the magnitude of increase is lower for poultry than for the milk. Outstanding dynamics of TPC for milk can be noticed in Figure 2.

TPC for milk started to rise from its lowest point of 1856.53 m rubles in 2013 to achieve an increase of more than 15 times in 2018. Other commodities in scope of the Russian import ban (potatoes, beef and veal, pig meat, poultry) showed a slight increase after 2014 and decrease in the period of 2015–2016. Such significant increase in value of TPC might show the de-facto monetary support to Russian producers of milk by Russian consumers conveyed through the price channel.

However, before identifying the source of higher TPC values, it is important to understand the split of the TPC between different regions of Russia.

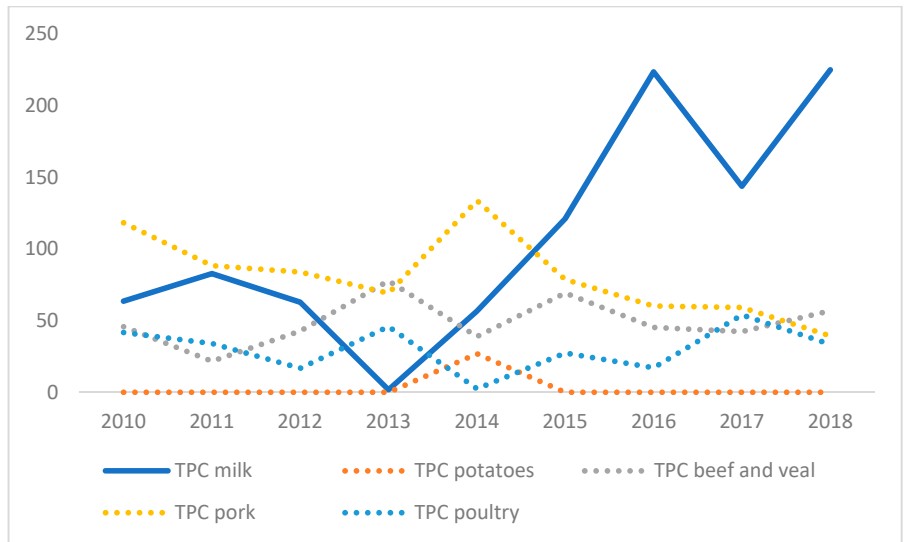

**Figure 2.** Transfers to producers from consumers (TPC) for different commodities in the Russian Federation, in Russia. Source: OECD [31].

Development of calculated regional TPCs for federal districts of the Russian Federation are shown in the Figure 3.

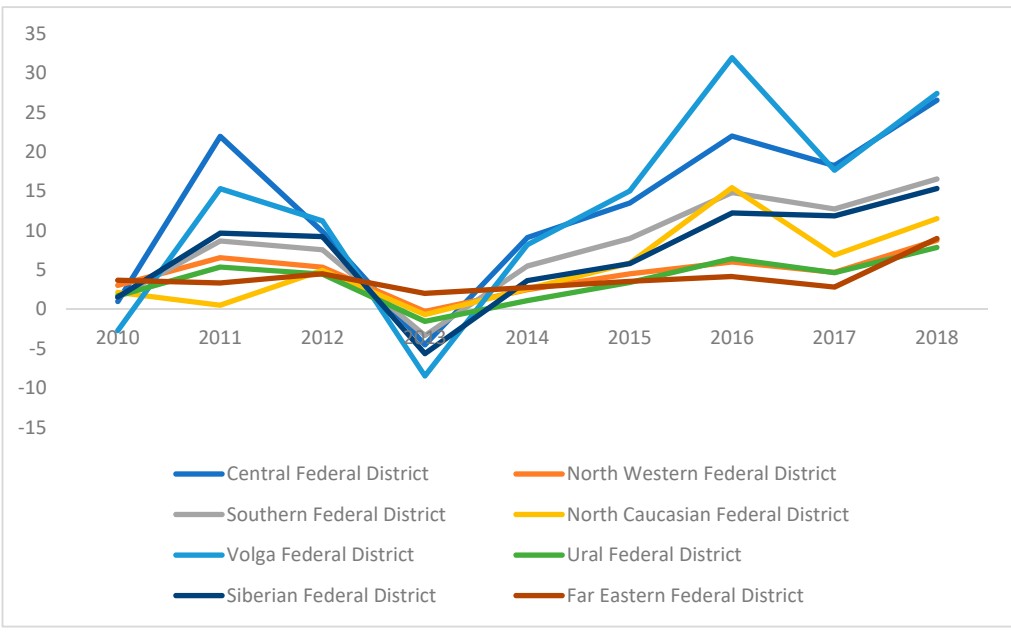

**Figure 3.** Similar trends in transfers to producers from consumers of milk (TPC) for federal districts of the Russian Federation, adjusted for inflation, billion RUB. Correlation coefficients among all pairs of districts, except for pairs involving the Far Eastern and North Caucasian districts, are greater than 0.9. Source: own calculation based on Rosstat [33] and OECD [31].

In the case of milk in 2013 (before import ban), transfers to producers from consumers (or TPC) were close to zero on federal level as per the OECD estimate. At the same time, recalculation methodology gives negative TPC for all federal districts except Far Eastern, while the main driver is the difference between regional producer prices and reference price. Volga Federal District was the

region with lowest TPC among regions, with value of 8468 m rubles, while North Western Federal District was the region with the closest to zero TPC of 273.61 m rubles. In 2015, the next year after the import ban was introduced, all regions have shown change from negative to positive values of TPC, and Central Federal District was the leader with positive TPC of 9083 m rubles. Ural Federal District became a region with lowest TPC of 1088 m rubles, while North Western Federal District showed TPC of 2488 m rubles.

The Russian import ban has resulted in increased transfers from consumers to producers due to increase in producers' price. This effect is evident across all Russian regions and without exception. Effectively, the import ban provided milk producers with monetary support due to differences between actual price on the regional market and possible border price, i.e., price of imported products. Moreover, the increases in TPC were observed in all the regions after 2014, and the main driver of the increase is growing producers' price. Producers' prices of milk in Russian regions are shown on the Table 8.

**Table 8.** Producers' prices of milk and price growth factors for federal districts of the Russian Federation, first differences, adjusted for inflation, RUB per ton. Source: Rosstat [33].

| Year | CFD | NWFD | SFD | NCFD | VFD | UFD | SIBFD | FEFD |
|---|---|---|---|---|---|---|---|---|
| 2011 | 566 | 4424 | 2356 | 2892 | −771 | 2233 | 1965 | 1476 |
| 2012 | −260 | −1975 | −245 | −41 | 2509 | 14 | 67 | 627 |
| 2013 | 3602 | 1808 | 1216 | 1006 | 2515 | 2566 | 1467 | 1546 |
| 2014 | 1357 | 3026 | 1826 | 3697 | 1311 | 2306 | 1504 | 2174 |
| 2015 | 1000 | 1439 | 1774 | 999 | 1869 | 627 | 1824 | 395 |
| 2016 | 3645 | 3039 | 1261 | 3058 | 6198 | 2943 | 2599 | 1646 |
| 2017 | 1329 | 320 | 3949 | 542 | −3464 | 1857 | 1597 | 9091 |
| 2018 | −1700 | −257 | −2617 | −1104 | −2656 | −1304 | −2533 | 1949 |
| | | | | Price growth factors | | | | |
| 2018/2010 | 1.695 | 2.028 | 1.709 | 1.943 | 1.606 | 2.029 | 1.683 | 2.603 |
| 2014/2010 | 1.383 | 1.633 | 1.384 | 1.645 | 1.449 | 1.652 | 1.402 | 1.494 |
| 2018/2014 | 1.225 | 1.242 | 1.235 | 1.181 | 1.108 | 1.229 | 1.200 | 1.743 |

All Russian regions experienced growth in producers' prices of milk during observed period of 2010–2016. However, the assessment of average year-on-year growth rates shows that the price growth changed after 2014 (Table 9).

**Table 9.** Changes in producers' prices of milk for federal districts of the Russian Federation, adjusted for inflation, chain index. Source: own calculation based on Rosstat [33].

| Year | CFD | NWFD | SFD | NCFD | VFD | UFD | SIBFD | FEFD |
|---|---|---|---|---|---|---|---|---|
| 2011/2010 | 4% | 38% | 18% | 25% | −6% | 20% | 16% | 13% |
| 2012/2011 | −2% | −12% | −2% | 0% | 22% | 0% | 0% | 5% |
| 2013/2012 | 26% | 13% | 8% | 7% | 18% | 19% | 10% | 11% |
| 2014/2013 | 8% | 19% | 11% | 24% | 8% | 15% | 9% | 14% |
| 2015/2014 | 5% | 8% | 10% | 5% | 10% | 3% | 10% | 2% |
| 2016/2015 | 18% | 15% | 6% | 15% | 31% | 16% | 13% | 9% |
| 2017/2016 | 6% | 1% | 18% | 2% | −13% | 9% | 7% | 46% |
| 2018/2017 | −7% | −1% | −10% | −5% | −12% | −6% | −11% | 7% |
| Average YoY 2010–2014 | 9% | 15% | 9% | 14% | 10% | 14% | 9% | 11% |
| Average YoY 2015–2018 | 6% | 6% | 6% | 4% | 4% | 6% | 5% | 16% |

All Russian regions, except Ural and Far Eastern federal districts, yielded an increase of producers' price of milk after 2014 in the range from 2.2% (Volga federal district) to 14.07% (Far Eastern federal district). The highest increase is observed in the Far Eastern federal district, which was preceded by an already high price rise in 2010–2014 (in average by 11.45% annually). The chain index (presented in Table 9) shows that producer prices of milk had the highest pace of increase in 2014,

the year in which the Russian import ban was introduced. At the same time, many federal districts experienced two-digit growth in 2015, 2016, and in 2017. It is interesting to notice that prices were decreasing in 2018, while milk production has significantly increased in 2017 (this will be touched upon later in the text).

All in all, there is an evidence of price increase and, in turn, increase in transfers from consumers to producers (Table 10). Pace of this increase has been different for federal districts of the Russian Federation, and the effect has been fading since 2017; however, changes in transfers have influenced the structure of support for agricultural producers in Russia. Unlike neighboring countries (mainly, member states of the European Union), structure of support has been changed, and the burden of support has been moved from taxpayers to consumers. This issue should be investigated more closely.

**Table 10.** Changes in calculated TPCs for federal districts of the Russian Federation, inflation adjusted, million RUB. Source: own calculation based on Rosstat [33].

| Year | CFD | NWFD | SFD | NCFD | VFD | UFD | SIBFD | FEFD |
|---|---|---|---|---|---|---|---|---|
| Average 2010–2013 | 7059.59 | 3657.47 | 3606.79 | 1721.00 | 3833.33 | 2485.78 | 3685.90 | 3368.90 |
| Average 2014–2017 | 15,693.37 | 4410.91 | 10,495.19 | 7678.46 | 18,191.58 | 3882.58 | 8371.73 | 3307.64 |
| Absolute change of average | 8633.78 | 753.44 | 6888.41 | 5957.46 | 14,358.25 | 1396.79 | 4685.83 | −61.26 |
| % change of average | 122.30% | 20.60% | 190.98% | 346.16% | 374.56% | 56.19% | 127.13% | −1.82% |

## 4. Discussion

According to the recalculations of federal TPC to regional level as per proposed methodology, all regions, except Far Eastern Federal District, yielded an increase in four-year averages after 2013, i.e., after introduction of the import ban. The four-year average of federal TPC for milk has grown from 52,738 m rubles in 2010–2013 to 136,175 m rubles in 2014–2017, which equates to 83,436 m rubles in absolute increase. The region with highest absolute increase of TPC is Volga Federal District, where it has grown from 3833 m rubles in 2010–2013 to 18,191 m rubles in 2014–2017.

After the import ban introduction, four-year average value for milk TPC changed by 158.21% on federal level, while the percentage changes on regional level differed from federal level. The most notable recorded change was in the case of Volga Federal District (374%), while the smallest change was in the Far Eastern Federal Region, where the four-year average has declined by 1.82%. Results for the Far Eastern Federal District may be supported by the fact that supply chains in this region are in the position of lowest dependence on the imports from the countries under import ban, while, for example, Volga Federal District is one of the regions with highest import from the countries under the scope of the import ban.

Transfers to producers from consumers is one of the types of support for agricultural producers. Specifically, TPC falls into the category of support based on commodity output. As can be seen from the OECD data (Figure 4), this type of support has become the biggest vehicle of support for agricultural producers in Russia after 2014. Share of support based on output has increased from 52.26% in 2013 (before import ban) to 71.47% in 2018, together with a decrease in share of payments based on input use from 37.23% in 2013 to 21.69% in 2018. Support for agriculture has shifted to support through price channel, rather than to support from state budget, in contrast with the European Union, where decoupled support has become the main support type for producers (Figure 5).

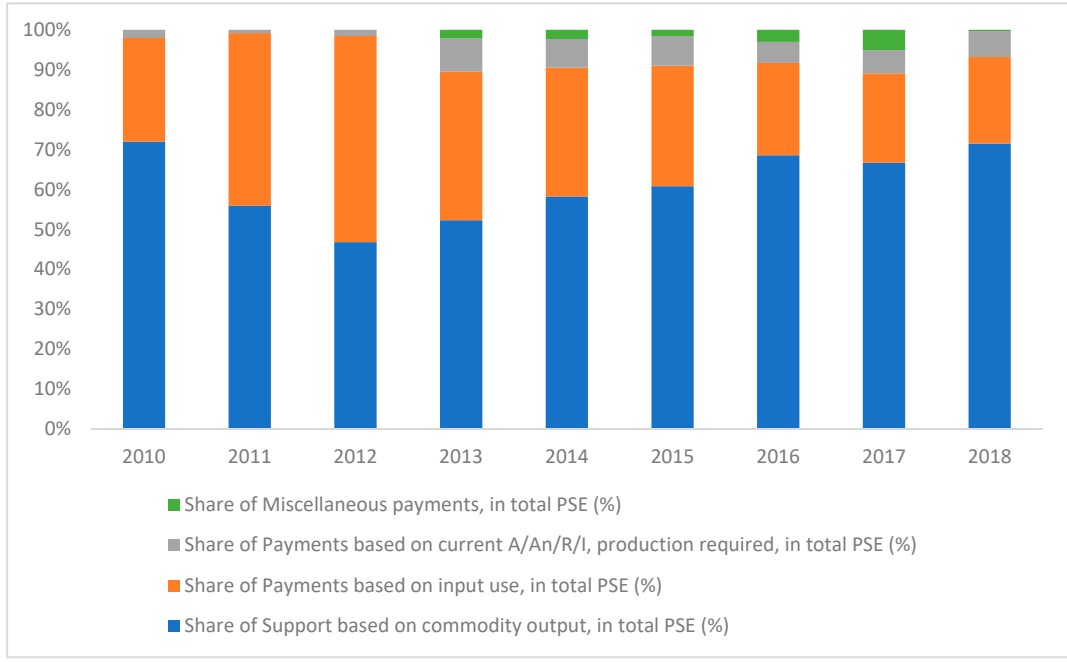

**Figure 4.** Structure of support for agricultural producers in Russia in 2010–2018. Source: OECD [31].

The European Union, the main exporter of milk and dairy products to Russia before the import ban, has changed the structure of agricultural support during 2010–2018 in a different direction, as opposed to Russia. The biggest part of support is attributed to payments based on non-current area, animal number, receipts, or income, with production not required (Figure 5). At the same time, share of payments based on current area, animal number, receipts, or income, with production required, has increased from 19.67% in 2010 to 25.42% in 2018.

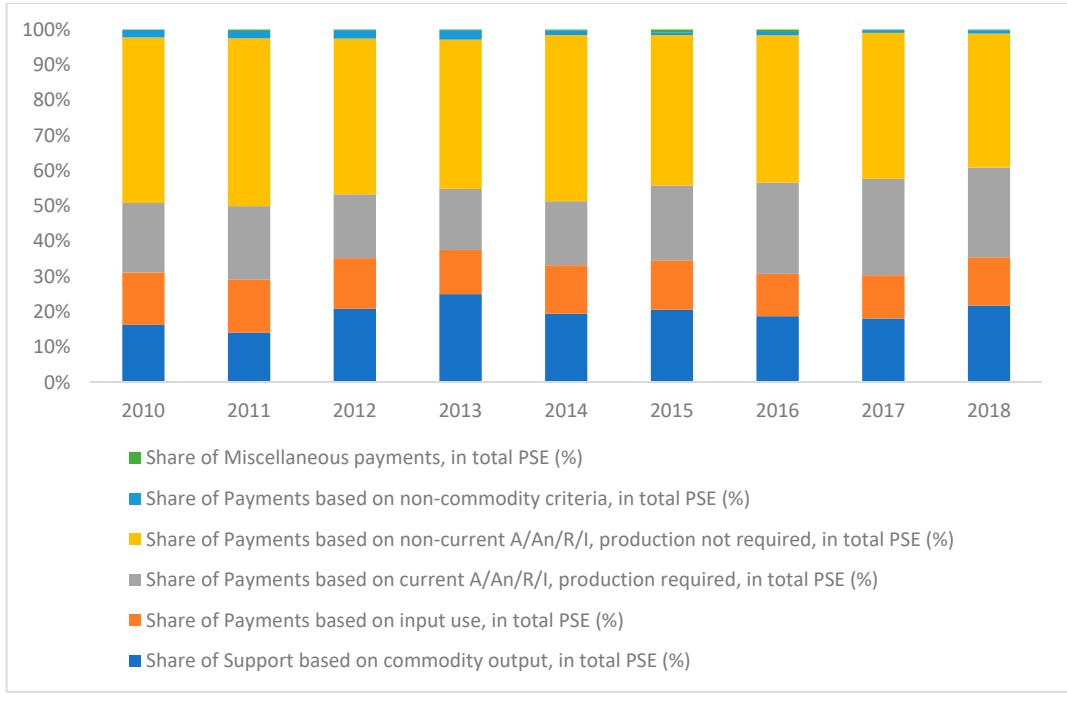

**Figure 5.** Structure of support for agricultural producers in EU28 in 2010–2018. Source: OECD [31].

Increased transfers from consumers to producers should have resulted in additional benefits to producers. It should also have acted as an additional incentive to invest in production facilities, which should then lead to increase in production. Interestingly, milk production has been growing since 2016 in Russia, as well as declining since 2014 in the European Union, which is shown in Figure 6.

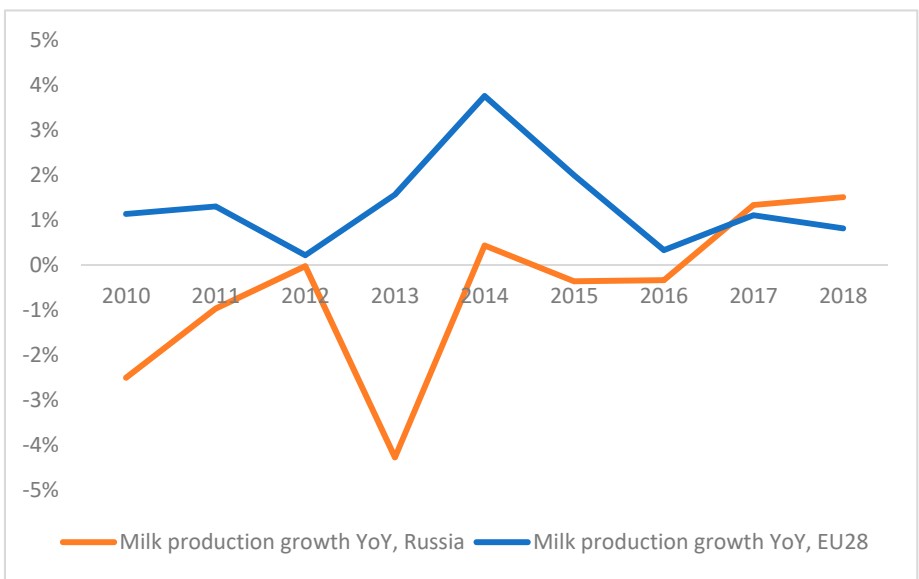

**Figure 6.** Milk production growth year over year in Russia and EU28. Source: own calculation based on OECD [31].

Production volumes in dairy industry in Russia started to grow in 2017, two years after the Russian import ban. As was mentioned by Semenova and Shumeiko [3], the Russian dairy industry can be characterized by lag between demand and production, which is reflected in the production growth of 2014–2018.

Petrick and Götz [9] have argued that current practices of subsidizing Russian dairy producers do not result in herd growth. Comparing the structure of producers' support in Russia and EU28, it is possible to see that subsidies are not the main channel of support for agricultural producers. Share of support that is not connected to output (payments based on input use, payments based on current area and animal number) has been declining in Russia since 2012 (53.24% of total support) to 28.19% in 2018. Therefore, investments in the dairy industry, in general terms, and herd growth, in particular, might be more closely associated with support based on commodity output, or support through price channel.

As one of the results of the import ban, Russia has chosen a different way, in contrast with the European Union. As there is a clear trend to decoupled payments for agricultural producers in the European Union, Russian agricultural producers receive more support from consumers through marginally higher prices. Comparing production growth in EU28 and Russia, one can notice that this approach has resulted in higher growth of milk production in Russia, rather than in the European Union. From this point of view, there is little incentive for the Russian government to lift the import ban. Moreover, as was shown by Banse et al. [21], the scenario with no import ban for years 2017–2030 shows that milk production in Russia will decrease by an average of 0.8% annually.

As has been shown previously, there are significant differences among federal districts of Russia in terms of price and production changes after the Russian import ban. The current study attempted to show the direction of these differences. In comparison to many other neighboring countries, the Russian Federation is the unique case of geographical and economic heterogeneity of the country. There is a definite opportunity for further studies of changes in individual members of each federal districts, in order to deepen current state of knowledge on import ban effects.

As in many cases during 20th century, economic sanctions have not achieved their purpose, but rather had less expected externalities. Instead of influencing on international policy of Russia, it has evoked the import ban and has led to strengthening of the Russian dairy industry. Therefore, it is difficult to consider that the Russian import ban will be lifted soon, as there is little economic incentive to do so.

## 5. Conclusions

The dairy and milk market were quite different in federal districts of the Russian Federation since the beginning of the 1990s, and it has undergone significant changes after introduction of the Russian import ban in August 2014. From the statistical standpoint, it is possible to reject the null hypothesis of equal mean for production volume in most of the pairs of federal districts. However, in the case of producer prices, the null of equal means cannot be rejected for most of the region pairs. This suggests significant differences in production volumes, but similar tendencies in producer prices. The only exception in this finding is the Far Eastern Federal District, for which the null of equal means can be rejected in pairs with Volga, Ural, and Siberian. The Far Eastern District is unique in many aspects, as can be seen from prices, production volumes, and TPC data, and it seems to become more disconnected from the rest of the country.

Recalculation of federal CSE estimation to regional level demonstrates that, in the case of the import ban of the Russian Federation, aggregate data might not show a detailed and complete picture of the policy effect. Due to significant differences between regions, there is an evidence of different pace in changes the period after 2014. Nevertheless, almost all of the regions experienced the increase in transfers to producers from consumers due to increase in producers' price, while reference price did not experience the changes in the same pace. This led to the situation in which the negative TPC in 2013 was replaced by the positive TPCs in all the regions in 2014–2016, which effectively means an increase in support of producers by consumers. Producers received the unique opportunity to charge consumers with higher prices, as competition has been lowered by the import ban. From this perspective, producers of milk might be considered as net beneficiaries of the import ban. In terms of milk and dairy products, the results also suggest that cost of the ban is being carried by consumers.

Unique opportunity for dairy producers in Russia has been used to increase production capacity, reflected by growth of milk production after 2017. Paradoxically, the import ban has strengthened the dairy industry in Russia, while consumers paid with support for agricultural producers instead of a state budget. Share of support related to commodity output, including market price support due to higher prices, has grown and has put agricultural producers into the position of lower dependency on subsidies.

Results of the study show that, currently, the Russian Government has little incentive to lift the import ban for milk and dairy products, as Russian consumers have shown willingness to pay for the dairy products at higher prices. This might not be a voluntary decision; however, it has formed a new equilibrium on the milk market.

The changes in transfers between producers and consumers constitute clear evidence of the import ban effect on the milk market in Russia. The format of the market was changed by the import ban and the state having not reversed it to the previous equilibrium but, rather, having formed a new one. It is a fact that the introduction of the Russian import ban was expected to increase self-sufficiency of Russia in terms of food, which is largely discussed and accepted in current research literature (see, e.g., References [25,26]). The increased transfers to producers from consumers of milk show more evident monetary part of the cost of self-sufficiency.

Further research in the field of changes on the milk market in Russia after the import ban might focus on the analysis of price cointegration in individual regions which, in turn, form federal districts. This analysis would provide more detailed view on the topic, as well as reveal several features that are overlooked on more aggregated level.

**Author Contributions:** Conceptualization, M.K. and L.S.; methodology, M.K.; validation, L.S.; formal analysis, M.K. and L.S.; writing—original draft preparation, M.K.; writing—review and editing, L.S.; visualization, M.K.; supervision, L.S.; funding acquisition, M.K. and L.S. All authors have read and agreed to the published version of the manuscript.

**Funding:** This research was funded by the grant of Internal Grant Agency (IGA) of Faculty of Economics and Management, Czech University of Life Sciences, project 2020A0015 entitled "Price changes on agrifood markets in post-Soviet countries under Russian import ban and sanctions", and by the grant of National Agency for Agricultural Research (NAZV) entitled "Dualita v českém zemědělství: výhoda nebo nevýhoda pro zemědělství nové generace?" (QK1920398).

**Acknowledgments:** This research was supported by the grant of Internal Grant Agency (IGA) of Faculty of Economics and Management, Czech University of Life Sciences, project 2020A0015 entitled "Price changes on agrifood markets in post-Soviet countries under Russian import ban and sanctions", and by the grant of National Agency for Agricultural Research (NAZV) entitled "Dualita v českém zemědělství: výhoda nebo nevýhoda pro zemědělství nové generace?" (QK1920398).

**Conflicts of Interest:** The authors declare no conflict of interest. The funders had no role in the design of the study; in the collection, analyses, or interpretation of data; in the writing of the manuscript, or in the decision to publish the results.

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
