# Peer review of "Trade Sanctions and Agriculture Support in Milk and Dairy Industry: Case of Russia"

_sustainability, doi:10.3390/su122410325_

Round 1
Reviewer 1 Report
The paper is potentially interesting, but it needs further improvement before publications.
The paper employs OECD methods to estimate the costs of trade ban for consumers and producers.
The paper analyses the regional heterogeneity of Russian milk production using rather aggregated data.
THe authors employs simple t test to check the regional heterogeneity for a pooled sample. I think this approach may lead to misleading results due to several reasons.First, obvious problem here is that pooled data may hide dynamics over time in Russian milk porduction. I understand that you have only 7 federal districts which is not enough to analyse seriously the regional heterogeneity, but pooling together data is not necessarily good solution.
From story telling point of view it would be better to show the market share of districts over time to get some insights for the reader about the context.
Similary argument can be apply for price statistics in Table 3.
My major problem is here that at the federal districts level of aggregation is fully meaningless to employ any basic statistics to show the regional heterogeneity, it would require much more disaggregated data.
If you like to estimate CSE at federal districts level is O.K. for me, but it does not require any "pseudo scientific" approach to show the regional heterogeneity of milk production and consumption.
I suggest to replace this part of the text focusing more on Russian context during the analysed period using some simple graphs on market share on production and consumption sides and producer and consumer prices.
The presentation of results in table 7-10 is poor. The reader obviously will lost in too many numbers in the Tables. You have to think about how to translate these numbers into graph to present much more visibly your results, e.g. using line graphs. Please consults with any datavisualization textbooks e.g. Evergreen: Effective data visualization, or Camoes: Data at work.
The quality of Figure 1 is also poor. E.g. you can omit gridlines, and decrease the number of zeros on the Y axis using other scales etc. You have similar problems with your other charts. You have to pay more attention on the desing of your graphs.
Finally, I am not convinced that simple CSE calculations are justify to publish this paper in Sustainability.
Reviewer 2 Report
The paper presents an analysis of the impact on domestic production of the Russian ban on the import of dairy products. The evaluation is performed by the estimation of the transfer from consumers to producers among different districts of the Russian Federation.
The introduction is well organized and allows the reader to correctly frame the topic. The methodology, results, and discussion sections show many limitations.
Point by point comments is now presented.
Line 53: when they are explicitly cited, it is suggested to put the number of bibliographic references next to the name of the authors. Adapt the entire text according to this tip.
Line 72: Authors should better clarify the level of self-sufficiency of the dairy sector in 2014. Are there expectations to improve self-sufficiency? How do you explain the general trend in herd reduction? Is it due to the increase in the productivity of cows higher than consumption increase?
Methodology section: Authors should present some descriptive statistics of districts about some kye variables.
Lines 193-200: the analysis of price variation over time cannot be performed applying ANOVA to raw data. Values should be at least de-trended.
Equation (2) and (3): why authors present the calculation of CSE if they analyze TPC in the rest of the paper? I suggest studying only TPC values.
Line 218: the authors should explain at this point how "reference price" is defined.
Table 2: This analysis is not useful. We know that countries and regions are different because are different in terms of geography, citizens, etc.: why compare absolute values? It would be more interesting to test if there are differences in self-sufficiency (%), milk consumption per-capita, production per-capita, etc.
Tables 6, 7, 8, 10: To improve the readability, I would remove the decimal.
Table 7: data over time should be at least deflated to be comparable.
Table 7: authors should at least try to explain why in 2013 TPC values are mainly negative.
Line 330: North-Western Federal District had the lowest value only in absolute terms. The lowest value is SIBFD while the highest is FEFD.
Table 8: the use of nominal price may hide a general trend in price. Actually, the global price decreased from 2014, but the author should take this into account.
Table 8: The last three lines are not explained in the table caption.
Table 9: A comparison with the global trend may show that Russian prices were linked to the global trend until 2014 and after become uncorrelated.
Table 10: which index has been used to adjust the inflation?
Discussion: The comparison of the absolute number of TPC among districts has a small interest. Results may gain interest if the TPC value is calculated in relative terms: - TPC per-capita - TPC per ton of Milk - TPC per farm - Incidence of TPC over gross receipt, etc.
Line 396: maybe the authors mean "exporter" instead of "importer".
Figure 4: I would use a bar chart instead of a line one. If authors want to use a line chart, I suggest presenting data on the production index.
Lines 431-434 and 462-468: Conclusions are weak and based on the results of other authors. What are the costs of society paying additional prices? Maybe consumers that lose food variety and experiencing higher prices are not so happy about it. In theory, the ban creates a net loss for the society that can not be measured only looking to milk production. You state on the abstract that dairy farms gain in terms of efficiency, but any evaluation has been performed.
Round 2
Reviewer 1 Report
The authors have addressed some comments, but others is still remaining unsolved.
I think that my major concerns is still valid, the data are too aggregated for the empirical analysis.
The quality of graphs have improved a bit, but new graphs also deserves further improvements.
You think that quality of graphs is not important issue at all and it is just a crazy reviewer's hobby. However, you have to think about from your potential reader point of view.
For example, there is a long time debate about pie/donut charts. Your Figure 1 is a good example, why pie chart is inefficient tool for data visualization.
https://www.storytellingwithdata.com/blog/2014/06/alternatives-to-pies
When you have more than 4 variables for times series analysis to create good graph is a challenging issue (Figure 3-4). Perhaps better solution to show separately each variables and place them under each other. I know that in Excel it is not an easy job.
It is also useful to add variable name directly to the end of your line using data label option in excel.
Reviewer 2 Report
The paper improved on some points, but some aspects are still unanswered.
Taking into account the response to the comments, there are the following additional requests.
Point 7: Table 2: This analysis is not useful. We know that countries and regions are different because are different in terms of geography, citizens, etc.: why compare absolute values? It would be more interesting to test if there are differences in self-sufficiency (%), milk consumption per-capita, production per-capita, etc.
Response 7: We compare absolute values as we try to take a look on the problem from the perspective of agricultural support to producers from the state budget. Later in the paper we conclude, that structure of the support has been changed, when support from state budget has been effectively substituted by support via price. In general terms, consumers are paying additionally in order to support producers, while comparing to European Union (a counter-party of the import ban), producers receive less support from price differences. We believe this to be a key supporting for the fact, that Russian import ban is unlikely to be lifted soon, as state expenses to support agricultural producers have been substituted by transfers from consumers.
Reply 7: If the aim is to compare the support to producers from the state budget with the value of price support paid by consumers, the authors should perform this comparison. There is still no reason to verify if among districts there are differences in absolute values.
Point 19: Lines 431-434 and 462-468: Conclusions are weak and based on the results of other authors. What are the costs of society paying additional prices? Maybe consumers that lose food variety and experiencing higher prices are not so happy about it. In theory, the ban creates a net loss for the society that cannot be measured only looking to milk production. You state on the abstract that dairy farms gain in terms of efficiency, but any evaluation has been performed.
Response 19: Efficiency of dairy farms has increased in monetary terms, as even with the constant level of production the increase in prices leads to increase in gross farm receipts.
Based on our findings (difference in agricultural support between Russian and European Union, main counter-party of the import ban) we conclude, that there is little incentive for Russian government to lift the import ban anytime soon, as the ban has helped to substitute the state budget expenses (to support agricultural producers) by additional expense from consumers (increases in transfers to producers from consumers after the import ban). At the same time, the paper focuses on the changes from the side of milk producers, rather than consumers or producers of other commodities.
Reply 19: The authors misuse the term efficiency. An improvement in efficiency implies an increase in physical production with the same production factors used or a reduction in the use of factors with the same output. The increase in gross agricultural income at the same price cannot be described as an improvement in efficiency. At the same time, the authors point to the divestment of direct support systems by the government, with negative effects on revenues that are not mentioned.
Round 3
Reviewer 1 Report
Probably we would never agreed regarding the appropriateness of your dataset for your research papers. The quality of your graphs is still poor. There is still room for decluttering e.g. elimination of too many zeros and % marks from the axis.
Author Response
Graphs have been corrected as per proposed changes. Zeros on axis have been eliminated, gridlines omitted in all graphs.